# Semi-conducting 2D rectangles with tunable length via uniaxial living crystallization-driven self-assembly of homopolymer

Sanghee Yang [1], Sung-Yun Kang [1] & Tae-Lim Choi [1✉]

Semi-conducting two-dimensional (2D) nanoobjects, prepared by self-assembly of conjugated polymers, are promising materials for optoelectronic applications. However, no examples of self-assembled semi-conducting 2D nanosheets whose lengths and aspect ratios are controlled at the same time have been reported. Herein, we successfully prepared uniform semi-conducting 2D sheets using a conjugated poly(cyclopentenylene vinylene) homopolymer and its block copolymer by blending and heating. Using these as 2D seeds, living crystallization-driven self-assembly (CDSA) was achieved by adding the homopolymer as a unimer. Interestingly, unlike typical 2D CDSA examples showing radial growth, this homopolymer assembled only in one direction. Owing to this uniaxial growth, the lengths of the 2D nanosheets could be precisely tuned from 1.5 to 8.8 μm with narrow dispersity according to the unimer-to-seed ratio. We also studied the growth kinetics of the living 2D CDSA and confirmed first-order kinetics. Subsequently, we prepared several 2D block comicelles (BCMs), including penta-BCMs in a one-shot method.

[1] Department of Chemistry, Seoul National University, Seoul 08826, Korea. ✉email: tlc@snu.ac.kr

Two-dimensional (2D) organic/polymeric nanosheets have attracted tremendous attention due to their unique properties arising from their ultrathin and flat morphology[1–5]. For instance, semi-conducting 2D materials such as graphene show optoelectronic properties that have been applied in the fields of sensors[4], electronic transfer platforms[5,6], and photovoltaic cells[7,8]. To prepare such functional 2D nanostructures by solution process, self-assembly of semi-crystalline block copolymers (BCPs) containing a solubilizing amorphous block is one of the most powerful and facile way to achieve various morphologies including rectangles[9], hexagons[10,11], diamonds[12,13], and squares[14,15]. Their core crystalline blocks can be varied from nonconjugated polymers, such as poly-(ferrocenyl dimethylsilane) (PFS)[9,10,16], poly(lactic acid) (PLLA)[12,13,17], poly($\varepsilon$-caprolactone) (PCL)[11,18], and polyethylene (PE)[19], to conjugated ones, including poly(3-hexylthiophene) (P3HT)[20,21], and poly(para-phenylenevinylene) (PPV)[14].

Since the properties of these nanomaterials are size-dependent[22–25], there have been numerous efforts to develop strategies including simple blending and heating to modulate their sizes, shapes, and dimensions[26–28]. In particular, the most powerful and widely utilized method is crystallization-driven self-assembly (CDSA), creating uniform one-dimensional (1D) and 2D nanomaterials[9–12,14,18]. Taking advantage of their living polymerization-like processes, preparing even more complex block comicelles is possible[29,30]. However, unlike inorganic or small organic molecules[31,32], semi-crystalline polymers undergo chain folding during CDSA. In the case of 2D nanosheets, this leads to radial growth of 2D lamellae, such that the aspect ratio can be predicted[10,33], but the current technology does not allow for precise control of the length. In particular, for semi-conducting 2D nanosheets, there is another limitation in that the strong $\pi$–$\pi$ interaction of the conjugated polymers lowers the solubility of the 2D nanostructures, resulting in irregular aggregation and uncontrolled self-assembly. While some problems may be solved by synthesizing BCPs containing nonconjugated shell blocks[14], these insulating blocks further limit the potential of the 2D nanosheets as electronic materials.

Despite the current success in controlling 2D nanostructures, understanding the 2D crystallization process in solution is still in its infancy. This study is of great importance as it allows the design of new polymers, broadening the scope of 2D nanostructures[34–36]. To this end, the Manners group recently reported a growth kinetic study on the formation of 1D nanofibers[37]. They observed that the conformational effect from the amorphous shell block of the BCPs disturbed the self-assembly process so that the kinetics were more complicated than those of analogous living polymerization and assembly of small molecules[38–40]. In order to obtain a clear kinetic study, controlled self-assembly of homopolymers is required to eliminate this conformational effect of BCPs. However, CDSA from homopolymers without a stabilizing shell block is extremely challenging, with only a few successes reported by the Manners group, who used a novel strategy of introducing a charged end group into homopolymers, thereby inducing electrostatic repulsion[10,41]. Due to these limitations, the kinetics of polymer self-assembly, especially quantitative 2D growth, have not been studied yet.

Regarding the formation of semi-conducting 2D nanosheets by self-assembly of a simple homopolymer, we recently reported two successful cases using conjugated poly(cyclopentenylene vinylene) (PCPV) homopolymers containing side chains of fluorene and bulky substituents such as neohexyl or silyl groups (Fig. 1a)[42,43]. We were able to control both the height and the aspect ratio of the resulting 2D nanosheets, but living CDSA was unsuccessful; instead, introducing another PCPV block as a shell, as in BCP, allowed for the living CDSA of 1D nanofibers with

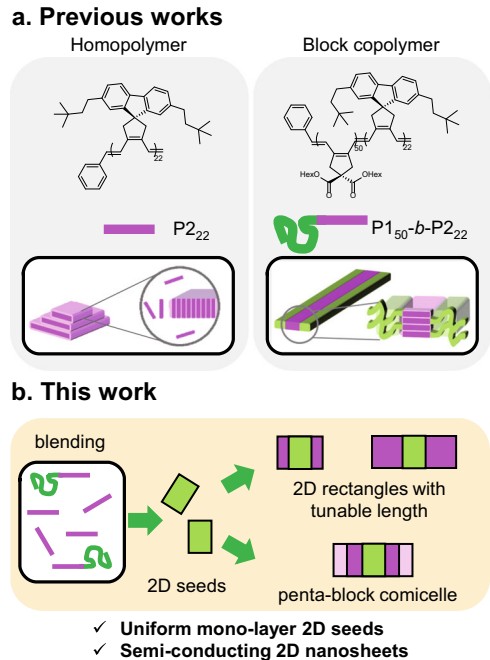

**Fig. 1 Successful strategy to obtain uniform semi-conducting 2D rectangles by living CDSA. a** Structures of the PCPV homopolymer and BCP and illustrations of the resulting 2D nanosheets and 1D nanofibers from the PCPVs. The green color in polymer chains indicates a P1 shell block, and the pink color in polymer chains indicates a P2 core block. **b** Illustration of strategies for producing uniform 2D rectangles by living CDSA.

tunable widths and lengths (Fig. 1a)[44]. Inspired by earlier studies, we envisioned that blending of the BCP and homopolymers might provide an excellent method for precisely controlled self-assembly of semi-conducting polymers. Herein, we demonstrate the successful formation of uniform semi-conducting 2D nanosheets without stacking using a blending strategy (Fig. 1b)[9,45]. From these initial 2D seeds, living 2D CDSA by a seeded-growth mechanism was possible by adding the homopolymer as a unimer. Intriguingly, the homopolymer assembled onto the 2D seeds uniaxially, resulting in the formation of length and area-controlled 2D nanorectangles with sharp edges. Based on real-time imaging of the living CDSA, kinetic studies on the uniaxial growth of 2D assembly revealed a first-order rate law, exactly following the living polymerization-like kinetics. Finally, we succeeded in forming complex block comicelle structures using unimers of various molecular weights.

## Results

**Formation of uniform 2D nanosheets by blending polymers.** We previously reported that conjugated PCPV homopolymers containing because of their uniform orthorhombic crystalline arrays. For example, the semi-crystalline P2$_{22}$ ($M_n$ = 9.01 kDa ($Đ$ = 1.13)) homopolymer directly assembled into 2D rectangles in chloroform but any control of the structure was impossible because of its low solubility (Fig. 1a)[42]. We then expanded this moiety to BCP microstructures composed of identical core-forming P2$_{22}$ blocks and solubilizing P1$_{50}$ shell blocks (P1$_{50}$-b-P2$_{22}$, $M_n$ = 38.6 kDa ($Đ$ = 1.10)) (Fig. 1a). This modification stabilized the P2 crystalline block, enabling precise control of widths and lengths via living CDSA, but a limitation was that only 1D nanofibers were prepared by this method[44]. Based on these

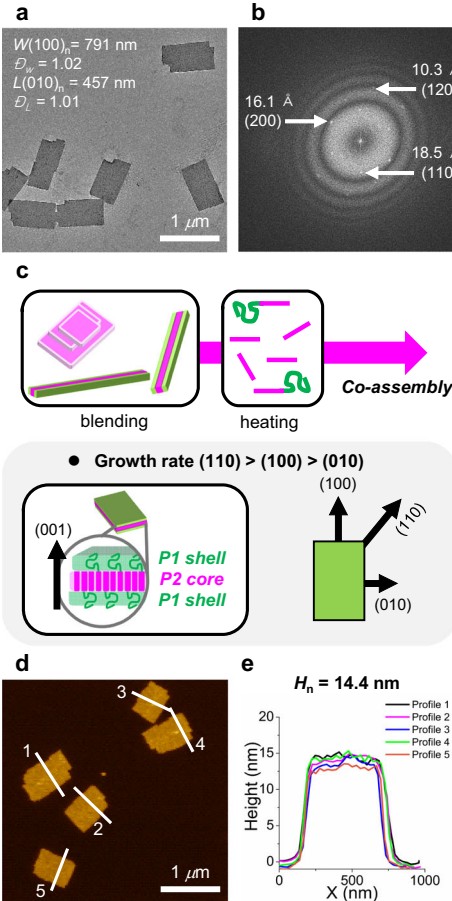

**Fig. 2 Characterization of uniform monolayer 2D seeds. a** TEM image of uniform 2D seeds generated by a heating and aging method using 2:1 blend ratio of $P1_{50}$-$b$-$P2_{22}$ and $P2_{22}$ homopolymer. Numbers in the image indicate the average width ($W_n$), length ($L_n$), and dispersity ($Đ_W$ or $Đ_L$). **b** FFT pattern from the HR-TEM image of the single 2D seed showing three main $d$-spacings of 10.3, 18.5, and 16.1 Å and their corresponding (hkl) planes. **c** Schematic representation of the formation of 2D seeds and their proposed detailed structures. Also included is their crystal array and observed crystal growth rates. **d** AFM image of 2D seeds and **e** height profile along the white lines shown in the AFM image, indicating the average height ($H_n$).

earlier investigations, we attempted the formation of uniform 2D nanosheets by blending $P2_{22}$ and $P1_{50}$-$b$-$P2_{22}$ to achieve controlled co-assembly. Partially introduced P1 shell blocks in the blends should stabilize the main crystal array of the $P2_{22}$ homopolymer, overcoming the solubility issue of 2D nanorectangles in solution (Fig. 1b). To achieve this co-assembly, we used a simple heating and aging method using the blends in chloroform and screened various conditions by changing the mass ratios of the two polymers, concentration, and temperature (Supplementary Figs. 1–4).

After many optimizations, excellent co-assembly was achieved by heating a blend solution of $P1_{50}$-$b$-$P2_{22}$ and $P2_{22}$ at a ratio of 2:1 (or molar ratio of 1:2) in 0.5 g/L chloroform at 50 °C for 1 h. After cooling to 25 °C and aging for 3 days, transmission electron microscopy (TEM) imaging showed uniform monolayers of 2D rectangles with an aspect ratio of 1.73 and an average angle of 92.9°, demonstrating excellent self-assembly via a self-seeding mechanism (Fig. 2a and Supplementary Fig. 5). Furthermore, the resulting 2D seeds had a uniform width ($W_n$) of 790.7 (±104.8) nm, length ($L_n$) of 456.7 (±39.6) nm, and an area ($A_n$) of 0.36 (±0.063) μm² with very narrow length and area dispersity (<1.02).

Cryogenic TEM imaging by freezing a low concentration of 0.05 g/L in chloroform also confirmed that this 2D self-assembly occurred in solution and not by solvent evaporation (Supplementary Fig. 6). To gain insight into these co-assembled 2D seeds, we analyzed their electron diffraction patterns by fast Fourier transform (FFT) analysis from the high-resolution TEM (HR-TEM) image (Fig. 2b and Supplementary Fig. 7). The resulting diffraction analysis showed an orthorhombic crystal lattice with main $d$-spacing values of 10.3, 16.1, and 18.5 Å, which were identical to those of the reported P2 homopolymer. This supports the conclusion that the blend of two polymers having common P2 cores co-crystallized into 2D seeds (Fig. 2c)[42]. Interestingly, the longer sides of the 2D rectangles always coincided with the direction of the (100) plane of the crystalline array. During the aging process, polymer nucleation formed small nuclei first, which then grew in both directions with slightly faster growth along the (100) plane than the (010) plane of the crystalline P2 core (cf. (110) > (100) > (010)) (Fig. 2c and Supplementary Fig. 8). Finally, the uniform 2D seeds having rectangular shapes with two distinct, well-defined crystalline surfaces were formed. In addition, The fact that the P2 core block stands upon the surface along the (001) direction of the 2D crystal lattice implies that the P1 shell blocks from the BCP would occupy the top and bottom of the 2D seeds, thereby suppressing the multistacking problem frequently observed in the previous single P2 assembly and making the 2D seeds colloidally stable[11,42]. This orientation also affected the height of the 2D seeds, which was measured by AFM analysis. Their average height ($H_n$) was 14.4 ± 0.4 nm, which was 3 nm higher than that of the previous 2D rectangles[42] just from the P2 homopolymer without the additional P1 shell block (Fig. 2d, e, and Supplementary Fig. 9).

**Uniaxial living seeded growth of homopolymers onto 2D seeds.** With the uniform 2D seeds in hand, we investigated the possibility of CDSA via seeded growth to further control the area of 2D nanosheets. A solution of lower molecular weight $P2_{10}$ (unimer, $M_n$ = 5.0 kDa, $Đ$ = 1.15) in 10 g/L chloroform was added to a solution of the 2D seeds in 0.03 g/L chloroform ($A_n$ of 0.32 (±0.059) μm²) with various unimer-to-seed (U/S) mass ratios from 2 to 15. After optimizations, $P2_{10}$ successfully underwent CDSA to form uniform 2D rectangles, whose area increased linearly from 1.2 to 7.1 μm² ($Đ$ < 1.02) according to U/S ratios after 3 weeks of aging at −13 °C (Figs. 3a, b, Supplementary Fig. 10). From the TEM images of the resulting 2D rectangles, the central 2D seeds appear darker, making them easily distinguishable from the newly formed 2D sheets derived from the $P2_{10}$ unimer (Fig. 3a). To our surprise, unlike other 2D platelets that grew in radial (both terminal and lateral) directions relative to the seeds, these 2D rectangles from the $P2_{10}$ unimer grew only in one direction along the (010) plane of the 2D seeds. This uniaxial growth enabled us to control the $L_n$ (010) of the resulting 2D rectangles from 1.5 to 8.8 μm while maintaining the width ($W_n$) of the (100) direction (Fig. 3c and Supplementary Fig. 11, 12). Low-magnification TEM images show that the length dispersity was less than 1.03, indicating successful living 2D CDSA (Supplementary Fig. 10). Living CDSA was also qualitatively supported by DLS analysis, where the $D_h$ values in chloroform solution gradually increased from 454 nm to 4.9 μm as the U/S ratios increased (Fig. 2d). The height of 2D rectangles in the (001) direction by AFM analysis also revealed a distinct difference between the 2D seeds and the newly formed 2D sheets (ca. 14.4 ± 0.4 nm versus 7.8 ± 0.6 nm, respectively) due to a higher DP of the P2 blocks for the seeds (Fig. 3e and Supplementary Fig. 13). To understand this unique uniaxial growth of the 2D rectangles along the (010) direction of the seeds as opposed to the faster

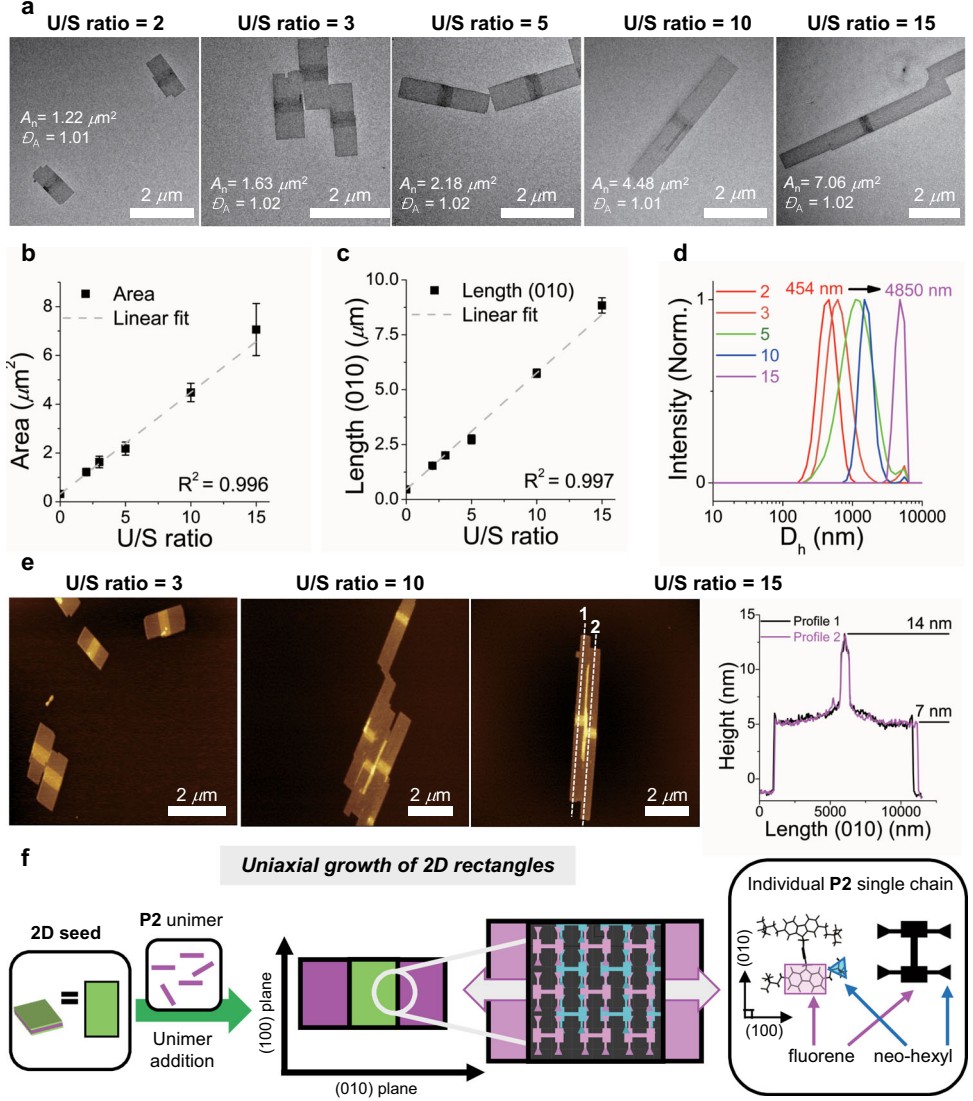

**Fig. 3 Uniaxial living 2D CDSA of P2 unimer. a** TEM images of length and area-controlled 2D rectangles with U/S ratios of 2, 3, 5, 10, and 15. Numbers in the images indicate the average area ($A_n$) and dispersity ($Đ_A$). Plots showing the linear dependence of **b** the area ($A_n$) and **c** the average length ($L_n$) in the (010) direction versus U/S ratios, demonstrating living 2D CDSA. Error bars indicate standard deviations ($\sigma$). **d** DLS profiles showing an increase in $D_h$ from U/S ratio of 2–15 after three weeks of aging. **e** AFM images of the resulting 2D rectangles prepared from U/S ratios of 3, 10, and 15. A height profile of the 2D rectangles with U/S ratio of 15, demonstrating a height difference (7.8 nm of the newly formed 2D sheets versus 14.4 nm of the original 2D seeds). Much darker middle blocks in TEM images and brighter middle blocks in AFM images indicate the 2D seeds. **f** Schematic representation of the living 2D CDSA via uniaxial seeded growth along the (010) direction. The 2D schematic illustration of resulting 2D rectangles is based on the interdigitating slip-stack packing model of P2 homopolymer with the simplified structure in *ab* plane (see Supplementary Fig. 14 for detail)[42].

growth in the (100) direction of the seed formation process, which might be under thermodynamic influence[46], we closely examined the orientation of the orthorhombic crystal lattice of the P2 homopolymer in the 2D seeds (Fig. 3f)[43]. Its (010) plane was occupied by rigid fluorene moieties of the P2 chains and would probably have much higher surface energy compared to the (100) plane exposing the neohexyl group (Supplementary Fig. 14). Therefore, during the elongation process, such distinct crystalline planes of the 2D seeds would allow the P2 unimers to kinetically crystallize onto the direction of higher surface energy, thereby leading to the preferential crystallization of unimers along the (010) direction. Similarly, in our previous finding, the 2D rectangular nanosheets from another PCPV homopolymer containing silyl groups also grew faster in the (010) direction than in the (100) direction with the 2D seeds having distinct crystalline surfaces[44]. In addition, the P2 chains containing *trans* alkenes

exclusively seemed to exhibit a fully extended conformation without chain folding in the 2D arrays, thereby maximizing the selective assembly of unimers to the seeds. This defect-free CDSA might have produced the resulting 2D sheets having sharp edges with a nearly perfect right angle[43].

**Growth kinetic studies on living 2D CDSA.** Most notably, this uniaxial living 2D CDSA is an excellent system for conducting kinetic studies of 2D assembly in solution. By real-time monitoring using TEM analysis, we conducted growth kinetics by adding P2$_{10}$ homopolymer with various U/S ratios under the aforementioned conditions (0.03 g/L seeds in chloroform solutions at −13 °C), and measuring increases in length, as described in Fig. 4a (Supplementary Table 1, Supplementary Figs. 15–19). As expected, the higher U/S ratio and concentration of unimer,

$[U]_0$, led to faster elongation. Interestingly, plotting $L_n$ growth *versus* time fitted very well with the first-order kinetic function of $[U]_0$, with $R^2$ values greater than 0.991, similar to conventional living polymerization (Table 1, Entries 1–5, Supplementary Eq. 1, Supplementary Fig. 20)[37]. The rate constants (k') were also consistent within experimental errors regardless of the U/S ratio (Supplementary Table 2). Using an alternation method, the reaction order of [U], which was calculated from the initial rates by analyzing the $L_n$ increase in the early stages, was 0.964 (±0.085) (Fig. 4b, Supplementary Eq. 2, Supplementary Fig. 21, Supplementary Table 3). This further confirmed the first-order kinetics of the living 2D CDSA of the P2 homopolymer. This is an interesting result because previous studies on living 1D CDSA of P1-*b*-P2 or PFS BCPs showed significant deviation from first-order kinetics[37,44]. The main difference in the present study is

presumably due to the homopolymer microstructure of unimers, unlike BCPs with shell blocks in the previous cases. Therefore, the P2 homopolymer has negligible conformational effects, due to the absence of a shell block. Finally, P2, which does not even undergo chain folding, directly forms crystalline arrays, making this 2D CDSA analogous to ideal crystallization, similar to the living supramolecular polymerization of small organic molecules such as porphyrin derivatives[38–40,47].

To explore the effect of temperature on CDSA, we again measured the rates at four different temperatures: 0, −10, −20, and −25 °C. After first-order fitting, the initial rate constant k' was found to increase from $3.0 \times 10^{-3}$ to $1.1 \times 10^{-2}$ as the temperature decreased from 0 to −25 °C (Supplementary Table 4 and Supplementary Fig. 22). These values allowed us to generate an Eyring plot to determine the activation enthalpy and entropy for the seeded-growth process, and they were negative values of −31.7 kJ/mol and −384 J/K•mol, respectively, indicating faster growth at lower temperatures (Table 1, Entries 6–9, Fig. 4c, Supplementary Eq. 3, Supplementary Table 5, Supplementary Fig. 23)[37]. Furthermore, another kinetic study was performed with longer P2 unimers of DP 13 (5.6 kDa ($Đ = 1.18$)) and 15 (6.1 kDa ($Đ = 1.13$)). Since the longer P2 unimers showing high crystallinity easily underwent self-nucleation at low temperatures, all the kinetic experiments were performed at 25 °C for proper comparison (Supplementary Table 6). Another first-order fitting of $[U]_0$ provided initial rate constants k' of $1.3 \times 10^{-4}$, $1.5 \times 10^{-3}$, and $2.5 \times 10^{-3}$ (h$^{-1}$) for P2 unimers of DP 10, 13, and 15, respectively (Table 1, Entries 10–12, Fig. 4d, Supplementary Fig. 24). This 20 times faster growth of $P2_{15}$ than $P2_{10}$ is likely due to the higher crystallinity of $P2_{15}$. Similar to our previous report on $P2$[42], the average height of the new 2D sheets increased from 7.8 to 10.4 nm according to the DP of P2 (Supplementary Fig. 25). With the longer P2 unimers, we were able to conduct living 2D CDSAs with various U/S ratios from 1 to 10 at 25 °C, providing tunable $L_n$ of the resulting 2D rectangles from 0.64 to 5.5 μm for $P2_{13}$ and from 0.80 to 3.3 μm for $P2_{15}$, respectively. (Supplementary Figs. 26, 27).

**Formation of symmetric penta-block comicelles.** Intriguingly, the uniaxial living 2D CDSA could prepare more complex multi-block comicelles (BCM) via a series of seeded growth from various P2 unimers[30,48,49]. Analogous to block copolymerization via living polymerization, the sequential addition of $P2_{10}$ and $P2_{15}$ unimers provided length and height controlled penta-BCMs along the (010) direction (unimers in 10 g/L chloroform, U/S ratio of 3, [2D seeds] = 0.03 g/L in chloroform, Fig. 5a). By changing the addition order, two types of symmetric penta-BCMs, A($P2_{15}$)-B($P2_{10}$)-S(seed)-B-A and B-A-S-A-B, were generated with uniform length and narrow dispersity (Fig. 5b,

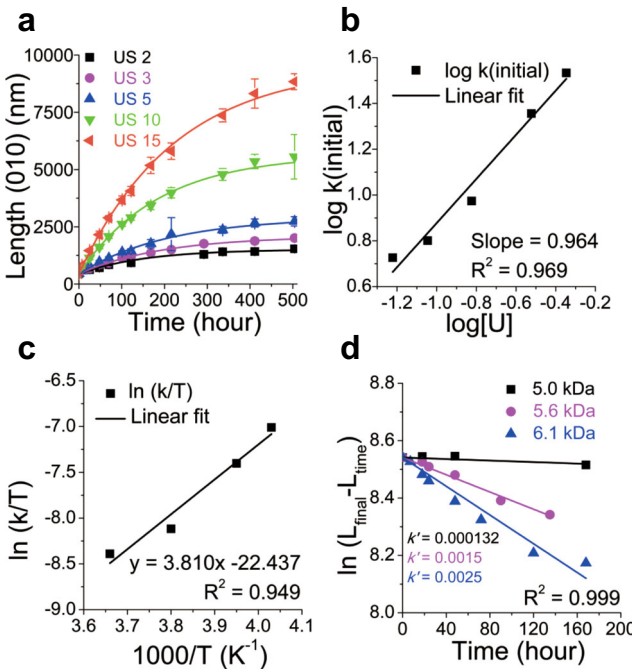

**Fig. 4 Growth kinetic studies on living 2D CDSA.** Plots showing **a** Lengths ($L_n$) of 2D rectangles over time with various U/S ratios from 2 to 15 (monitored for 3 weeks). Error bars indicate standard deviations ($\sigma$). **b** initial reaction rates *versus* unimer concentration, [U], to confirm first-order kinetics. **c** Eyring plot for the rate constants, k', extracted from seeded-growth experiments at varying temperatures. **d** Plot of $ln(L_{final} - L(time))$ versus time from the first-order growth of various unimers with different DPs at 25 °C.

| Table 1 Kinetic data for 2D CDSA experiments with different conditions: rate constants (k') with standard errors and $R^2$. | | | | | | |
|---|---|---|---|---|---|---|
| Entry | Unimer(kDa) | Agingtemperature (°C) | U/Sratio | k'(h$^{-1}$) | Error(h$^{-1}$) | $R^2$ |
| 1 | 5.0 | −13 | 2 | $6.5 \times 10^{-3}$ | $6.1 \times 10^{-4}$ | 0.991 |
| 2 | | | 3 | $5.1 \times 10^{-3}$ | $4.0 \times 10^{-4}$ | 0.993 |
| 3 | | | 5 | $4.9 \times 10^{-3}$ | $4.2 \times 10^{-4}$ | 0.991 |
| 4 | | | 10 | $5.4 \times 10^{-3}$ | $4.6 \times 10^{-5}$ | 0.995 |
| 5 | | | 15 | $4.4 \times 10^{-3}$ | $2.5 \times 10^{-5}$ | 0.997 |
| 6 | 5.0 | 0 | 10 | $3.0 \times 10^{-3}$ | $8.4 \times 10^{-5}$ | 0.999 |
| 7 | | −10 | | $3.8 \times 10^{-3}$ | $1.4 \times 10^{-4}$ | 0.999 |
| 8 | | −20 | | $7.3 \times 10^{-3}$ | $2.2 \times 10^{-4}$ | 0.999 |
| 9 | | −25 | | $1.1 \times 10^{-2}$ | $2.7 \times 10^{-4}$ | 0.999 |
| 10 | 5.0 | 25 | 10 | $1.3 \times 10^{-4}$ | $4.3 \times 10^{-5}$ | 0.999 |
| 11 | 5.6 | | | $1.5 \times 10^{-3}$ | $5.9 \times 10^{-5}$ | 0.999 |
| 12 | 6.1 | | | $2.5 \times 10^{-3}$ | $1.4 \times 10^{-4}$ | 0.999 |

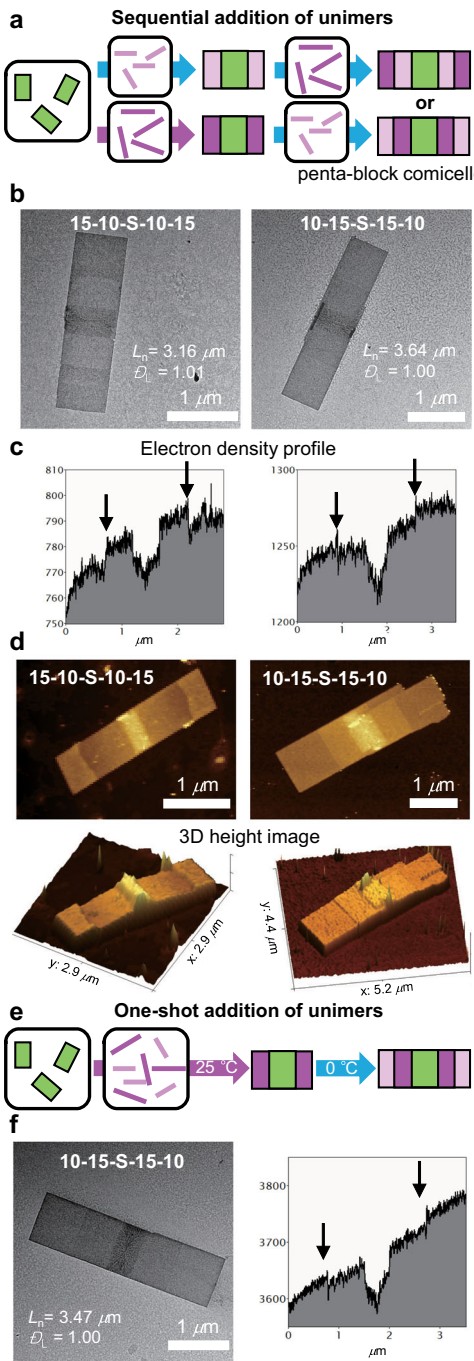

**Fig. 5 Successful formation of symmetric penta-BCMs. a** Schemes for the preparation of complex 2D BCMs by sequential addition of two unimers ($P2_{10}$ and $P2_{15}$). By changing the order of the unimer addition, two types of symmetric penta-BCMs were obtained. The 2D nanosheets of longer $P2_{15}$ unimers are shown in dark pink, and those of $P2_{10}$ unimers are shown in light pink. **b** TEM images, **c** electron density profiles, and **d** 2D, and 3D height AFM images of the penta-BCMs showing clear distinctions of the two types of penta-BCMs prepared by the sequential addition. **e** Scheme for the preparation of one 2D BCM by one-shot addition of two unimers. **f** TEM image and electron density profile of the penta-BCM obtained by one-shot living CDSA. Numbers in the images indicate the average length ($L_n$) and dispersity ($Đ_L$).

Supplementary Figs. 28, 29). A clear distinction in contrast is observed in the TEM images as the A block of $P2_{15}$ appears darker due to its higher electron density than the B block of $P2_{10}$ (Fig. 5c). Furthermore, AFM analysis also confirms the blocky

structure of penta-BCMs, showing another clear difference in the height of 2D sheets (Fig. 5d and Supplementary Fig. 30). We even attempted a more challenging but simple one-shot BCM formation by adding two P2 unimers to 2D seeds at the same time with each U/S ratio of 3 at 25 °C[50]. Since more crystalline $P2_{15}$ grew much faster than $P2_{10}$, the longer $P2_{15}$ preferentially assembled on the 2D seeds. Then, lowering the temperature to 0 °C initiated the self-assembly of the shorter $P2_{10}$ via the seeded-growth mechanism, resulting in the formation of the same B-A-S-A-B penta-BCM as that prepared by sequential addition (Fig. 5e, Supplementary Figs. 30, 31). This result demonstrates an excellent example of this new strategy that could be used to construct complex nanostructures based on a deep understanding of the 2D assembly process.

Since these precisely controlled 2D rectangles were composed of fluorescent conjugated PCPVs, they were visible under super-resolution structured illumination microscopy (SR-SIM) without additional dye. Interestingly, 2D seeds in the middle showed much higher fluorescence than the rest of the 2D sheets freshly formed from P2 unimers, indicating that the longer P2 emits stronger light (Supplementary Fig. 32). In addition, a video of the micelle solution recorded by confocal laser scanning microscopy (CLSM) shows a persistent shape and fluorescence stability without decomposition and photobleaching (Supplementary Video 1).

## Discussion

In conclusion, we successfully demonstrated the formation of uniform semi-conducting 2D rectangles having sharp edges from a semi-crystalline conjugated homopolymer by the uniaxial seeded-growth approach. This intriguing direction-selective assembly allowed us to control the length of 2D rectangles for the first time with narrow dispersity through 2D CDSA. Taking advantage of this uniaxial growth from a homopolymer, 2D growth kinetic studies revealed that the homopolymer self-assembly followed ideal first-order kinetics, similar to living polymerization. This result indicates that the polymer self-assembly follows ideal crystallization because disturbing elements, such as the conformational effect of a shell block or back-bone chain folding, were eliminated. Lastly, 2D CDSA produced several complex but well-controlled penta-BCMs using various sizes of P2. Ultimately, we succeeded in a one-shot BCM formation based on an understanding of the growth kinetics, which provided an excellent guideline for optimizing self-assembly conditions. These precisely controlled uniform fluorescent 2D nanostructures would have great potential for optoelectronic applications.

## Methods

**Polymerization procedure**. A 4 mL sized screw-cap vial with septum was flame dried and charged with a monomer and a magnetic bar. The vial was purged with $Ar_{(g)}$ three times, and degassed anhydrous THF was added ($[M1]_0 = 0.5$ M or $[M2]_0 = 0.1$ M). After the $Ar_{(g)}$-purged G3 catalyst in the other 4 mL vial was dissolved in 50 μL THF, the solution was rapidly injected to the monomer solution at 0 °C under vigorous stirring. After the complete conversion of M1 to P1 (or M2 to P2 for P2 homopolymer), for the block copolymer (BCP) formation, the second monomer (M2) was added ($[M2]_0 = 0.1$ M) to the vial at 0 °C. The reaction was quenched by excess ethyl vinyl ether (EVE) after the desired reaction time and precipitated in methanol at room temperature. The obtained purple solid was filtered and dried in vacuo. The monomer conversions were calculated from the $^1$H NMR spectra of the remained crude mixture. Both P1-*b*-P2 BCP and P2 homopolymer have been characterized separately in the reference[42,44].

**Preparation of blends of $P1_{50}$-*b*-$P2_{22}$ and $P2_{22}$**. Each polymer ($P1_{50}$-*b*-$P2_{22}$ and $P2_{22}$) was dissolved in 0.5 g/L chloroform (A total volume of blends was more than 0.5 mL in 4 mL vial). Without aging, two solutions were mixed at room temp. with various $P1_{50}$-*b*-$P2_{22}$ and $P2_{22}$ with a ratio of 2:1 (or molar ratio of 1:2).

**Preparation of 2D seeds from the blend by heating and aging**. The blended solutions were sealed with a Teflon lined cap and heated at 50 °C for 1 h, followed by cooling down to 25 °C and aging for 3 days. The resulting 2D nanoparticles were observed by AFM and TEM imaging. In details, samples for TEM and AFM imaging were prepared by spin-coating one drop (*ca.* 10 mL) of the 2D seeds colloidal solution onto a carbon-coated copper grid (for TEM imaging) or freshly cleaved mica (for AFM imaging) (spinning rate = 3000 rpm for 30 secs). Device-related information is described in the Supplementary Information.

**Preparation of 2D rectangles with tunable length via living 2D CDSA of P2$_n$ unimers**. After diluting the 2D seeds solution to 0.03 g/L chloroform, a solution of lower molecular weight P2$_n$ (unimer, $M_n$ = 5.0–6.1 kDa, Đ < 1.18) in 10 g/L chloroform was added to the solution of the 2D seeds with various unimer-to-seed (U/S) mass ratios. Then the samples were aged at X °C (variable temperatures).

**Growth kinetic studies of the living 2D CDSA**. To conduct growth kinetic studies of the 2D assembly in solution, we monitored the living 2D assembly over aging time after adding the unimer solution to the seed solution. For each nanostructure, length, width, height, area, aspect ratio, and angle distributions were estimated from the TEM and AFM images manually using the ImageJ software package, which developed at the US National Institute of Health. For the statistical length analyses, more than 30 randomly picked objects were processed to determine the average values depending on the dataset. Every particle in each image was counted to reduce subjectivity.

## Data availability
The data that support the findings of this study, including the Supplementary Information, are available from the corresponding author upon request.

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

## Acknowledgements

This research was supported by the NRF, Korea through the following grants: Creative Research Initiative Grant and Postdoctoral Fellowship Program (Nurturing Next-generation Researchers, NRF grant no. 2020R1A6A3A01095653). We also thank the National Center for inter-University Research Facilities (NCIRF) at SNU for supporting CLSM and SR-SIM.

## Author contributions

S.Y. and T.-L.C. conceived the project. S.Y. and S.-Y.K. carried out the experiments under the supervision of T.-L.C. All authors interpreted the data and discussed the experimental results.

## Competing interests

The authors declare no competing interests.
