## [Peer Review File · Nature Communications]

REVIEWER COMMENTS

Reviewer #1 (Remarks to the Author):

This is an interesting study. I like the precise control over the formation of 2D nanosheet by the self-assembly of conjugated homopolymers. Indeed, as the authors mentioned, it is very difficult to control the self-assembly of conjugated homopolymers as the strong p-p interaction significantly lowers their solubility in solution, resulting in irregular aggregation and uncontrolled self-assembly. Impressively, the authors overcome this challenge. They achieved the control over CDSA of a conjugated homopolymer in one direction and thus obtained beautiful nanosheets with controlled aspect ratio and length. The manuscript is well written and easy to follow. I have only three remarks for the authors before the acceptance of this work for publication.

1. In Supplementary Fig. 2, with an increase in the mass of the homopolymer, the resultant nanosheets appeared more and more square. In addition, when the block copolymer dominated in the co-assembly, there were a lot of long and thin 1D ribbon-like structures in the TEM images. Were these 1D structures formed by the self-assembly of only the block copolymer or by the co-assembly? Why did the increase of the homopolymer amount lead to the formation of nanosheets with a more square shape during the co-assembly?

2. The authors claimed that the (010) plane of the seeds "was occupied by rigid fluorene moieties of the P2 chains, and would probably have higher surface energy compared to the (100) plane exposing the neohexyl group, thereby leading to the preferential crystallization of unimers along this direction". If this speculation is correct, why is the growth rate along the (100) direction faster than that along the (010) direction during the formation of the seeds (Figure 2c)? They look contradictory (as we can compare Figure 2c and Figure 3f). Please explain.

3. Figure 5a is confused. Does it mean that both of the two types of the penta-block micelles were formed in both of the methods, OR only one type was formed in a corresponding method, and which was formed in which method?

Reviewer #2 (Remarks to the Author):

In this manuscript, the author used 2D crystallization sheets based on conjugated poly(cyclopentenylene vinylene) homopolymer and its block copolymer to realize the precise control of the length of the formed 2D morphologies in long strip shape by CDSA approach. The assembly proceeded along the 2-dimensional seed and grew in only one direction, which were special and pioneering jobs in the field of controlled 2D self-assembly. I think this paper has very high quality for Nat. Commun., and it could be accepted after adding and correcting some contents.

1. The experiment result indicated that the growth of the 2D seeds along the (100) plane is faster growth than the (010) plane, but the formed morphologies after CDSA grew along the (010) plane. I think this interesting phenomenon need detailed explanation, and this is the key point to explain the origin of morphology. The author simply attributed it to the higher surface energy of the (010) plane, which was lack of sufficient experimental and theoretical evidence support. I suggest the authors may consider using molecular dynamic to simulate the intermolecular interaction which may be able to uncover a clear mechanism. In addition, the schematic representation of Fig. 3f was confused, and I can not understand the meaning of the representation of the rectangle, the triangle, the line and the colors in the packing mode, which were not mentioned in the figure legend.

In addition, the measurement position in Fig. 2e should be marked in Fig. 2d.

2. The word “Semi-conducting” in the title were not well embodied in the manuscript. Almost none work in this manuscript use the semi-conducting property of the morphologies. I think that the author could explore the corresponding job in the future, but I suggest that this word should just simply be replaced by the word “conjugated”.

3. Some data in SI need reasonable explanation. For example, from Fig S10b and S26d, we can find that the shoulder peak at about 590 nm got weaker and weaker as the U/S ratios increased, so what is the reason? And from Fig S27d, it seemed the shoulder peak unchanged when the 2D rectangles were built by P215 unimer, so why the UV-Vis spectra based on the longer P2 block showed the different changes? From Fig S26c and Fig 27c, some results in the DLS plot were not consistent with the U/S ratio, is these just measurement error?

Reviewer #3 (Remarks to the Author):

The paper by Tae-Lim Choi is an elegant contribution to the field of crystallisation driven self-assembly (CDSA). This is an exciting field however most advances do not use technologically relevant polymers and this is where this contribution represents a major contribution to the field. This work focussing on the challenging area of semi-conducting precision assembly and demonstrates a new approach for access to 2D structures rectangular in shape with tuneable and controllable length. The key advance is the ability to enable uniaxial control in such a controlled and living manner – with a core crystallizing polymer which is difficult to control. Importantly such conducting materials have potential impact when their 2d dimensions can be controlled.

My only major comment is that the one-shot approach is not really well described as to why it works and what they do. This is an important advance given the nature of what the authors demonstrate can be achieved but as written the figures and description do not present it in as a clear as manner as is required.

I also suggest the figure captions needs further detail to assist the reader in understanding the detail of the experiments.

The rationale for the uniaxial growth needs strengthening as the model in fig 3f is sound but does seem to be observational rather than insightful.

A more convincing 10-15-s-15-10 AFM image should be presented the current image could be 2 nanostructures overlaid. Also for clarity where the 3d height image is measured in figure 5 would be helpful.

Response to Reviewer's comments

Our comments are shown below in blue and revised texts in the manuscript and Supplementary Information are highlighted.

Reviewer #1 (Remarks to the Author):

This is an interesting study. I like the precise control over the formation of 2D nanosheet by the self-assembly of conjugated homopolymers. Indeed, as the authors mentioned, it is very difficult to control the self-assembly of conjugated homopolymers as the strong p-p interaction significantly lowers their solubility in solution, resulting in irregular aggregation and uncontrolled self-assembly. Impressively, the authors overcome this challenge. They achieved the control over CDSA of a conjugated homopolymer in one direction and thus obtained beautiful nanosheets with controlled aspect ratio and length. The manuscript is well written and easy to follow. I have only three remarks for the authors before the acceptance of this work for publication.

- Thank you for your positive feedback.

1. In Supplementary Fig. 2, with an increase in the mass of the homopolymer, the resultant nanosheets appeared more and more square. In addition, when the block copolymer dominated in the co-assembly, there were a lot of long and thin 1D ribbon-like structures in the TEM images. Were these 1D structures formed by the self-assembly of only the block copolymer or by the co-assembly? Why did the increase of the homopolymer amount lead to the formation of nanosheets with a more square shape during the co-assembly?

- As shown in **Fig. 1a**, multi-stacked 2D rectangles of **P2** homopolymer (ref. 42, *JACS*, **139**, 3082 (2017)) and 1D nanofibers of BCP (**P1-b-P2**) (ref. 44, *Chem. Sci.*, **11**, 8416 (2020)) have been reported. In the growth mechanism of the multi-stacked 2D rectangles, **P2** homopolymer assembles in both the (100) and (010) directions of its crystalline plane, while the BCP crystallizes first in the (100) direction, resulting in the micrometer-long 1D nanofibers along the (100) direction. Therefore, as the mass of the BCP increases during co-assembly, more 1D ribbon-like structures appear close to the 1D nanofibers of BCP (Please see Supplementary Fig. 2 for detail). Conversely, when the portions of **P2** homopolymer increases in the blend, 2D nanosheets from the **P2** homopolymer are created, but with suppression of the multi-stack due to the BCP minimizing the interaction between each sheet. If the portions of **P2** homopolymer becomes too high, then the formation of multi-stacked 2D nanosheets is dominant again.

- Regarding this comment, we added more explanation in the caption of Supplementary Fig. 2.

- Supplementary Fig. 2 "As shown in **Fig. 1a**, multi-stacked 2D rectangles of the homopolymer⁴ and 1D nanofibers of BCP⁵ have been reported. In the growth mechanisms of the multi-stacked 2D rectangles, **P2** homopolymer assembles in both the (100) and (010) directions of the crystalline plane, while BCP crystallizes first in the (100) direction, resulting in the micrometer-long 1D nanofibers. Therefore, as the mass of the BCP increases during co-assembly, more 1D ribbon-like structures appear close to the 1D nanofibers of BCP. Conversely, when the portions of **P2** homopolymer increases in the blend, 2D nanosheets from the **P2** homopolymer are created, but with suppression of the multi-stack due to the BCP minimizing the interaction between each sheet. If the portions of **P2** homopolymer becomes too high, then the formation of multi-stacked 2D nanosheets is dominant again."

2. The authors claimed that the (010) plane of the seeds "was occupied by rigid fluorene moieties of the P2 chains, and would probably have higher surface energy compared to the (100) plane exposing the neohexyl group, thereby leading to the preferential crystallization of unimers along this direction". If this speculation is correct, why is the growth rate along the (100) direction faster than that along the (010) direction during the formation of the seeds (Figure 2c)? They look contradictory (as we can compare Figure 2c and Figure 3f). Please explain.

- Thank you for the comment. This is a very valid point. The preferred growth direction of the seed formation and the elongation processes seems to be different depending on the crystallization mechanism of the polymers. First, to understand the self-assembly mechanism of 2D seeds, we performed the real-time TEM imaging of 2D seeds formation during the aging process and observed the growth of 2D seeds over time. In detail, after 30 min aging at 25 °C, polymer nucleation proceeded by spontaneous aggregation of polymer chains to form small homogenous nuclei. These primary nuclei (seeds) grew less directionally, so unimer mixtures (homopolymer+BCP) assembled in two-dimensional ways with slightly faster growth rate in the (100) direction than in the (010) direction, as observed in the self-assembly of each of the BCP and **P2** homopolymer (Please refer to the response to the comment 1). This seed formation step seems to be a thermodynamic process as recently reported by in situ TEM analysis, which showed that the formation of Au nuclei was rather reversible (ref. 46, *Science*, **371**, 498 (2021)). Also, we recently found that this relative rate of (100) vs (010) growths can be altered depending on the seed formation conditions that we are investigating more in detail and hopefully we will publish it soon. Finally, we formed the uniform large 2D seeds having rectangular shapes with two distinct crystalline surfaces in the (100) and (010) directions. This seed formation process followed a typical self-seeding mechanism for single crystals of the blends.

- Supplementary Fig. 8 **a**, The formation process of 2D seeds over aging time from 30 min (after heating) to 1 day. **b**, Schematic illustration of the 2D seeds formation with different crystal growth rates. From their rectangular shapes, we could infer that the different width and length values would be due to the difference in the crystal growth rate of each plane. Thus, one side (100) of the rectangular nanosheets would grow slightly faster than the other side (010), leading to the formation of anisotropic rectangular nanosheets.³
- On the contrary, the next 2D-CDSA process between the 2D seeds and the **P2** homopolymer (acts as an unimer) followed the common seeded-growth mechanism. As aforementioned, the final 2D seeds already have well-defined two crystalline planes with very different arrangements: the (100) plane of the seeds was occupied by neohexyl groups and their (010) plane was occupied by rigid fluorene moieties. Therefore, during the elongation process, such distinct crystalline planes would allow the **P2** unimer to crystallize selectively onto the direction of higher surface energy (the (010) plane). This kinetically controlled seed-to-unimer assembly would be the main reason for the uniaxial growth, as we speculated in the manuscript. Similarly, in our previous finding (ref. 43, *JACS*, **141**, 19138 (2019)), we also observed that the 2D nanosheets grew faster in the (010) direction (where fluorene is exposed) than in the (100) direction when the 2D seeds have distinct crystalline surfaces in those planes. In sum, the preferential growth direction of each process is depending on the different crystallization mechanisms.
- As a response to the comment, we added more explanation of the preferential growth directions on **page 4** in the manuscript.
- "During the aging process, polymer nucleation formed small nuclei first, which then grew in both directions with slightly faster growth along the (100) plane than the (010) plane of the crystalline **P2**

core (cf. (110) > (100) > (010)) (Fig. 2c and Supplementary Fig. 8). Finally, the uniform 2D seeds having rectangular shapes with two distinct, well-defined crystalline surfaces were formed.”

“To understand this unique uniaxial growth of the 2D rectangles along the (010) direction of the seeds as opposed to the faster growth in the (100) direction of the seed formation process which might be under thermodynamic influence, we closely examined the orientation of the orthorhombic crystal lattice of the **P2** homopolymer in the 2D seeds (Fig. 3f).⁴² Its (010) plane was occupied by rigid fluorene moieties of the **P2** chains and would probably have much higher surface energy compared to the (100) plane exposing the neohexyl group (Supplementary Fig. 14). Therefore, during the elongation process, such distinct crystalline planes of the 2D seeds would allow the **P2** unimers to kinetically crystallize onto the direction of higher surface energy, thereby leading to the preferential crystallization of unimers along the (010) direction. Similarly, in our previous finding,⁴³ the 2D rectangular nanosheets from another **PCPV** homopolymer containing silyl groups also grew faster in the (010) direction than in the (100) direction with the 2D seeds having distinct crystalline surfaces.”

3. Figure 5a is confused. Does it mean that both of the two types of the penta-block comicelles were formed in both of the methods, OR only one type was form in a corresponding method, and which was formed in which method?

We are sorry for the confusion about Fig 5a. Both of the two types of the penta-block comicelles (10-15-s-15-10 and 15-10-s-10-15) were formed by the sequential method; however, with the one-shot method, only 10-15-s-15-10 BCM was formed due to preferential assembly of the **P2₁₅** unimers with high crystallinity.

To make this point clear, we modified Fig. 5 to separate the schematic illustrations for the two methods (Figs. 5a and 5e) and added more explanation in the figure caption.

Fig. 5 | Successful formation of symmetric penta-BCMs. **a**, Schemes for the preparation of complex 2D BCMs by sequential addition of two unimers (**P2₁₀** and **P2₁₅**). By changing the order of the unimer addition, two types of symmetric penta-BCMs were obtained. **b**, TEM images, **c**, electron density profiles, and **d**, 2D, and 3D height AFM images of the penta-BCMs showing clear distinctions of the two types of penta-BCMs prepared by the sequential addition. **e**, Scheme for the preparation of one 2D BCM by one-shot addition of two unimers. **f**, TEM image and electron density profile of the penta-BCM obtained by one-shot living CDSA. Numbers in the images indicate the average length (L_n) and dispersity.

Reviewer #2 (Remarks to the Author):

In this manuscript, the author used 2D crystallization sheets based on conjugated poly(cyclopentenylene vinylene) homopolymer and its block copolymer to realize the precise control of the length of the formed 2D morphologies in long strip shape by CDSA approach. The assembly proceeded along the 2-dimensional seed and grew in only one direction, which were special and pioneering jobs in the field of controlled 2D self-assembly. I think this paper has very high quality for Nat. Commun., and it could be accepted after adding and correcting some contents.

Thank you for your positive feedback.

1-1. The experiment result indicated that the growth of the 2D seeds along the (100) plane is faster growth than the (010) plane, but the formed morphologies after CDSA grew along the (010) plane.

I think this interesting phenomenon need detailed explanation, and this is the key point to explain the origin of morphology. The author simply attributed it to the higher surface energy of the (010) plane, which was lack of sufficient experimental and theoretical evidence support. I suggest the authors may consider using molecular dynamic to simulate the intermolecular interaction which may be able to uncover a clear mechanism.

- Thank you for the comment. This is a very valid point. The preferred growth direction of the seed formation and the elongation processes seems to be different depending on the crystallization mechanism of the polymers. First, to understand the self-assembly mechanism of 2D seeds, we performed the real-time TEM imaging of 2D seeds formation during the aging process and observed the growth of 2D seeds over time. In detail, after 30 min aging at 25 °C, polymer nucleation proceeded by spontaneous aggregation of polymer chains to form small homogenous nuclei. These primary nuclei (seeds) grew less directionally, so unimer mixtures (homopolymer+BCP) assembled in two-dimensional ways with slightly faster growth rate in the (100) direction than in the (010) direction, as observed in the self-assembly of each of the BCP and **P2** homopolymer (Please refer to the response to the comment 1). This seed formation step seems to be a thermodynamic process as recently reported by in situ TEM analysis, which showed that the formation of Au nuclei was rather reversible (ref. 46, *Science*, **371**, 498 (2021)). Also, we recently found that this relative rate of (100) vs (010) growths can be altered depending on the seed formation conditions that we are investigating more in detail and hopefully we will publish it soon. Finally, we formed the uniform large 2D seeds having rectangular shapes with two distinct crystalline surfaces in the (100) and (010) directions. This seed formation process followed a typical self-seeding mechanism for single crystals of the blends.

Supplementary Fig. 8 a, The formation process of 2D seeds over aging time from 30 min (after heating) to 1 day. **b**, Schematic illustration of the 2D seeds formation with different crystal growth rates. From their rectangular shapes, we could infer that the different width and length values would be due to the difference in the crystal growth rate of each plane. Thus, one side (100) of the rectangular nanosheets would grow slightly faster than the other side (010), leading to the formation of anisotropic rectangular nanosheets.³

- On the contrary, the next 2D-CDSA process between the 2D seeds and the **P2** homopolymer (acts as an unimer) followed the common seeded-growth mechanism. As aforementioned, the final 2D seeds already have well-defined two crystalline planes with very different arrangements: the (100) plane of the seeds was occupied by neohexyl groups and their (010) plane was occupied by rigid fluorene moieties. Therefore, during the elongation process, such distinct crystalline planes would allow the **P2** unimer to crystallize selectively onto the direction of higher surface energy (the (010) plane). This kinetically controlled seed-to-unimer assembly would be the main reason for the uniaxial growth, as we speculated in the manuscript. Similarly, in our previous finding (ref. 43, *JACS*, **141**, 19138 (2019)), we also observed that the 2D nanosheets grew faster in the (010) direction (where fluorene is exposed) than in the (100) direction when the 2D seeds have distinct crystalline surfaces in those planes. In sum, the

preferential growth direction of each process is depending on the different crystallization mechanisms.

- In fact, we are further investigating the uniaxial growth of 2D rectangles. To account for relative growth rates in each direction, we are studying the effects of the solvent and molecular weight of polymers on the relative growth rates in the 2D-seed formation process. Furthermore, to understand the mechanism of uniaxial growth of 2D rectangles in 2D-CDSA process, we are also planning to do the co-work for the molecular dynamics of the assembly. Hopefully, we will report these works in the near future. Thank you for your great suggestion.
- As a response to the comment, we added more explanation of the preferential growth directions on **page 4** in the manuscript.

"During the aging process, polymer nucleation formed small nuclei first, which then grew in both directions with slightly faster growth along the (100) plane than the (010) plane of the crystalline **P2** core (cf. $(110) > (100) > (010)$) (Fig. 2c and Supplementary Fig. 8). Finally, the uniform 2D seeds having rectangular shapes with two distinct, well-defined crystalline surfaces were formed."

"To understand this unique uniaxial growth of the 2D rectangles along the (010) direction of the seeds as opposed to the faster growth in the (100) direction of the seed formation process which might be under thermodynamic influence,⁴⁶ we closely examined the orientation of the orthorhombic crystal lattice of the **P2** homopolymer in the 2D seeds (Fig. 3f).⁴² Its (010) plane was occupied by rigid fluorene moieties of the **P2** chains and would probably have much higher surface energy compared to the (100) plane exposing the neohexyl group (Supplementary Fig. 14). Therefore, during the elongation process, such distinct crystalline planes of the 2D seeds would allow the **P2** unimers to kinetically crystallize onto the direction of higher surface energy, thereby leading to the preferential crystallization of unimers along the (010) direction. Similarly, in our previous finding,⁴³ the 2D rectangular nanosheets from another **PCPV** homopolymer containing silyl groups also grew faster in the (010) direction than in the (100) direction with the 2D seeds having distinct crystalline surfaces."

1-2. In addition, the schematic representation of Fig. 3f was confused, and I cannot understand the meaning of the representation of the rectangle, the triangle, the line and the colors in the packing mode, which were not mentioned in the figure legend.

- We apologize for the lack of explanation for the schematic representation of **Fig. 3f**. The illustrations of the rectangles, triangles, and lines have been reported in our previous paper (ref. 42, *JACS*, **139**, 3082 (2017)) and represented the slip-stack packing of **P2** homopolymer. We intended to show which growth direction the fluorene and neo-hexyl groups are involved in with this representation.
- As a response to the comment, we changed **Fig. 3f** and added more explanation about the schematic representation of simplified **P2** structure in the caption and Supplementary Fig. 14.

Fig. 3 | f, Schematic representation of the living 2D CDSA via uniaxial seeded-growth along the (010) direction. "The 2D schematic illustration of resulting 2D rectangles is based on the interdigitating slip-stack packing model of **P2** homopolymer with the simplified structure in *ab* plane (see Supplementary Fig. 14 for detail).⁴²"

Supplementary Fig. 14 "The fluorene moieties on the **P2** polymer chains are described as rectangles and the neo-hexyl groups as triangles. In detail, the interdigitating slip-stack packing of **P2** chains are described as **c**, the low-magnified illustration of the 2D arrangements of **P2** chains on *ab*-plane. Based

on this 2D illustration, we propose that the neo-hexyl moiety is exposed along the (100) direction, and the fluorene moiety is exposed along the (010) direction. This difference would probably lead to different surface energy, resulting in the preferential crystallization along one direction."

1-3. In addition, the measurement position in Fig. 2e should be marked in Fig. 2d.

- Thank you for the comment. We represented the measurement positions in **Figs. 2d** and **2e** and also marked the positions in Supplementary Figs. 13, 25, 30, and 31.

Fig. 2 | "d, AFM image of 2D seeds and e, height profile along the white lines shown in the AFM image."

2. The word "Semi-conducting" in the title were not well embodied in the manuscript. Almost none work in this manuscript use the semi-conducting property of the morphologies. I think that the author could explore the corresponding job in the future, but I suggest that this word should just simply be replaced by the word "conjugated".

- Thanks for the suggestion. We agree that experiments highlighting the semi-conducting property of 2D rectangles are lacking in this manuscript. We are currently exploring their optoelectronic properties through collaboration with physicists. In addition, unlike other examples of 2D nanostructures that contain insulating shell blocks, our 2D rectangles consist of fully conjugated polymers. Moreover, as the bandgap energy of 1.97 eV of the 2D rectangles is within the range of semiconductor materials, we think these 2D nanostructures meet the semi-conducting materials criteria. Also, the term conjugated applies to molecules (conjugated polymers) but not to nanomaterials: conjugated nanoparticles do not seem to sound right.

3. Some data in SI need reasonable explanation. For example, from Fig S10b and S26d, we can find that the shoulder peak at about 590 nm got weaker and weaker as the U/S ratios increased, so what is the reason? And from Fig S27d, it seemed the shoulder peak unchanged when the 2D rectangles were built by P215 unimer, so why the UV-Vis spectra based on the longer P2 block showed the different changes? From Fig S26c and Fig 27c, some results in the DLS plot were not consistent with the U/S ratio, is these just measurement error?

- First, we apologize for not being more careful with writing caption in SI. As U/S ratios increased, more **P2** unimer was added, and the ratio of **P2** core block to **P1** shell block of the total solution increased. Therefore, if the nano-particulation has less influence on the UV-vis absorption, the relative intensity of

the conjugated backbone signal (from 400 nm to 600 nm) from both blocks' conjugated backbones) decreased compared to 313 nm derived only from fluorene moiety of **P2** block.

- We re-normalized all of the UV-vis spectra in Supplementary Figs. 10b, 26d, and 27d with 313 nm, and confirmed that UV-vis spectra generally followed the above tendency even after the additional dilution process for UV-vis absorption analysis.
- The hydrodynamic diameters (D_h) measured by DLS were the average sizes of the nanostructures in solution, assuming the shape of nanostructures as spherical particles. Therefore, the D_h values are just suitable for qualitative support for increasing the size of 2D rectangles. More accurate measurements are from AFM and TEM.
- In addition, we added a more detailed explanation to the manuscript captions and Supplementary Information and highlighted the modifications: "Supplementary Figs. 2, 3, 5, 6, 7, 8, 9, 10, 11, 13, 14, 24, 25, 26, 27, 28, 30, 31 and Supplementary Table 6."

Reviewer #3 (Remarks to the Author):

The paper by Tae-Lim Choi is an elegant contribution to the field of crystallisation driven self-assembly (CDSA). This is an exciting field however most advances do not use technologically relevant polymers and this is where this contribution represents a major contribution to the field. This work focussing on the challenging area of semi-conducting precision assembly and demonstrates a new approach for access to 2D structures rectangular in shape with tuneable and controllable length. The key advance is the ability to enable uniaxial control in such a controlled and living manner – with a core crystallizing polymer which is difficult to control. Importantly such conducting materials have potential impact when their 2d dimensions can be controlled.

- Thank you for your positive feedback.

1. My only major comment is that the one-shot approach is not really well described as to why it works and what they do. This is an important advance given the nature of what the authors demonstrate can be achieved but as written the figures and description do not present it in as a clear as manner as is required.

- Thank you for your comments. To make the representation of the one-shot approach clearer, we modified Fig. 5 by separating the schematic illustrations for the two methods (Figs. 5a and 5e) and added more explanation in the figure caption.

Fig. 5 | Successful formation of symmetric penta-BCMs. **a**, Schemes for the preparation of complex 2D BCMs by sequential addition of two unimers (**P2₁₀** and **P2₁₅**). By changing the order of the unimer addition, two types of symmetric penta-BCMs were obtained. **b**, TEM images, **c**, electron density profiles, and **d**, 2D, and 3D height AFM images of the penta-BCMs showing clear distinctions of the two types of penta-BCMs prepared by the sequential addition. **e**, Scheme for the preparation of one 2D BCM by one-shot addition of two unimers. **f**, TEM image and electron density profile of the penta-BCM obtained by one-shot living CDSA. Numbers in the images indicate the average length (L_n) and dispersity.

2. I also suggest the figure captions needs further detail to assist the reader in understanding the detail of the experiments.

- Thank you for the comment. We added a more detailed explanation to the manuscript captions and

Supplementary Information and highlighted the modifications: "Captions of supplementary Figs. 2, 3, 5, 6, 7, 8, 9, 10, 11, 13, 14, 24, 25, 26, 27, 28, 30, 31 and Supplementary Table 6."

3. The rationale for the uniaxial growth needs strengthening as the model in fig 3f is sound but does seem to be observational rather than insightful.

- We apologize for the lack of explanation for the schematic representation of Fig. 3f. The illustrations of the rectangles, triangles, and lines have been reported in our previous paper (ref. 42, *JACS*, **139**, 3082 (2017)) and represented the slip-stack packing of **P2** homopolymer. We intended to show which growth direction the fluorene and neo-hexyl groups are involved in with this representation.
- As a response to the comment, we changed **Fig. 3f** and added more explanation about the schematic representation of simplified **P2** structure in the caption and Supplementary Fig. 14.

Fig. 3 | f, Schematic representation of the living 2D CDSA via uniaxial seeded-growth along the (010) direction. "The 2D schematic illustration of resulting 2D rectangles is based on the interdigitating slip-stack packing model of **P2** homopolymer with the simplified structure in *ab* plane (see Supplementary Fig. 14 for detail).⁴²"

Supplementary Fig. 14 "The fluorene moieties on the **P2** polymer chains are described as rectangles and the neo-hexyl groups as triangles. In detail, the interdigitating slip-stack packing of **P2** chains are described as **c**, the low-magnified illustration of the 2D arrangements of **P2** chains on *ab*-plane. Based on this 2D illustration, we propose that the neo-hexyl moiety is exposed along the (100) direction, and the fluorene moiety is exposed along the (010) direction. This difference would probably lead to different surface energy, resulting in the preferential crystallization along one direction."

In addition, we have responded to the comment and added more explanation of the preferential growth directions on **page 4** in the manuscript (Please refer to the response to the comment 2 of Reviewer #1 and comment 1 of Reviewer #2).

"During the aging process, polymer nucleation formed small nuclei first, which then grew in both directions with slightly faster growth along the (100) plane than the (010) plane of the crystalline **P2** core (cf. (110) > (100) > (010)) (Fig. 2c and Supplementary Fig. 8). Finally, the uniform 2D seeds having rectangular shapes with two distinct, well-defined crystalline surfaces were formed."

"To understand this unique uniaxial growth of the 2D rectangles along the (010) direction of the seeds as opposed to the faster growth in the (100) direction of the seed formation process which might be under thermodynamic influence,⁴⁶ we closely examined the orientation of the orthorhombic crystal lattice of the **P2** homopolymer in the 2D seeds (Fig. 3f).⁴² Its (010) plane was occupied by rigid fluorene moieties of the **P2** chains and would probably have much higher surface energy compared to the (100) plane exposing the neo-hexyl group (Supplementary Fig. 14). Therefore, during the elongation process, such distinct crystalline planes of the 2D seeds would allow the **P2** unimers to kinetically crystallize onto the direction of higher surface energy, thereby leading to the preferential crystallization of unimers along the (010) direction. Similarly, in our previous finding,⁴³ the 2D rectangular nanosheets from another **PCPV** homopolymer containing silyl groups also grew faster in the (010) direction than in the (100) direction with the 2D seeds having distinct crystalline surfaces."

4. A more convincing 10-15-s-15-10 AFM image should be presented. the current image could be 2

nanostructures overlaid. Also for clarity where the 3d height image is measured in figure 5 would be helpful.

- Thank you for the comment. We changed the AFM image for 10-15-s-15-10 BCMs with clearer ones in Supplementary Figs. 30 and 31d and modified the 3D height images of **Fig. 5d**.

- Supplementary Fig. 30 "AFM images of **a**, A(P2₁₅)-B(P2₁₀)-S(seed)-B-A penta-BCM and **b**, B(P2₁₀)-A(P2₁₅)-S(seed)-A-B penta-BCM prepared by the sequential addition of two unimers with U/S ratios of 3 and their height profiles along the white lines shown in the AFM images. Both penta-BCMs clearly showed the height distinction in the height profiles of 2D nanosheets."
- Supplementary Fig. 31 "**d**, AFM images of the B-A-S-A-B type of penta-BCM and height profile along the white line shown in the AFM image."

- **Fig. 5** | "**d**, 2D and 3D height AFM images of the penta-BCMs showing clear distinctions of the two types of penta-BCMs prepared by the sequential addition."

REVIEWERS' COMMENTS

Reviewer #1 (Remarks to the Author):

The authors have revised the manuscript carefully and answered my questions. I have no more comments and recommend the publication of the current version.

Reviewer #2 (Remarks to the Author):

The authors have made a reasonable response and careful revision. I recommend the publication of this manuscript in Nature Communications.

Reviewer #3 (Remarks to the Author):

the authors have fully addressed my comments and those of the other reviewers and i recommend the paper to now be accepted.